# Synthesis, Characterization and Properties of Soybean Oil-Based Polyurethane

**DOI:** 10.3390/polym14112201

**Published:** 2022-05-28

**Authors:** Qi Xu, Jianwei Lin, Guichang Jiang

**Affiliations:** College of Light Industry Science and Engineering, Tianjin University of Science and Technology, Tianjin 300222, China; xqxq20222022@163.com (Q.X.); jeavi_lin@163.com (J.L.)

**Keywords:** epoxy soybean oil, soybean oil-based polyurethane, polyethylene glycol

## Abstract

At present, the consumption of polyurethane is huge in various industries. As a result, it has become a research hotspot to use environmentally friendly and renewable bio-based raw materials (instead of petroleum-based raw materials) to prepare polyurethane. In this paper, epoxy soybean oil (ESO) was used as raw material, and polyethylene glycol (PEG-600) was used for ring opening. Fourier transform infrared (FT-IR) and proton nuclear magnetic resonance (^1^H NMR) analysis proved that soybean oil-based polyester polyols was prepared. Soybean oil-based polyurethane (SPU) was synthesized by the reaction of the soybean oil-based polyol with isophorone diisocyanate (IPDI), so as to save energy and protect the environment. The properties of SPU films were adjusted by changing the R value (the molar ration of -NCO/-OH) and the film forming temperature. The chemical structure and properties of the SPU were characterized by FTIR, ^1^H NMR, gel permeation chromatography (GPC), scanning electron microscopy (SEM), thermogravimetric analysis (TGA), and differential scanning calorimetry (DSC). The results show that the mechanical strength, water contact angle, microphase separation degree, barrier property, and thermal stability of SPU films gradually increase, while the transparency, oxygen permeability coefficient and moisture permeability coefficient of SPU films gradually decrease with the increase of R value and film forming temperature.

## 1. Introduction

Polyurethane (PU) is a macromolecular polymer with a urethane structure (-NH-COO-) on the main chain. Generally speaking, firstly, isocyanate reacts with polyol to form a polyurethane prepolymer, and then a chain extender is added to continue the reaction to obtain polyurethane. Among them, the isocyanate and chain extender constitute the hard segment of polyurethane, and the polyol constitutes the soft segment of the polyurethane.

Polyurethane is a common polymer industrial material which is widely used in adhesives, coatings, rubber, and other fields. Due to its universal high temperature resistance, excellent mechanical properties, and good flexibility. Its application range is becoming wider and wider, and demand for it is increasing [1,2,3]. However, the raw material of polyurethane comes from non-renewable petroleum resources. In order to save resources and protect the environment, replacing petroleum-based raw materials with degradable and renewable bio-based raw materials has become an inevitable trend in the future development of the polyurethane industry [4,5,6,7,8,9].

With the intensive research on polyurethane, people have discovered that vegetable oil is a potential target for replacing petroleum raw materials [10,11,12]. The vegetable oil has a special molecular structure containing hydroxyl or epoxy groups [13,14]. The epoxy ring-opening method is commonly used in vegetable oil modification to prepare vegetable oil-based polyols and participate in the synthesis of polyurethane [15,16,17,18].

The vegetable oil selected in this paper is epoxy soybean oil (ESO). ESO cannot react directly with isophorone diisocyanate (IPDI), so it needs to be modified. Polyethylene glycol (PEG-600) was used as a ring-opening agent to ring-open ESO, and soybean oil-based polyol was prepared. Then, SPU was prepared by the reaction between this polyol and IPDI. We expect to prepare a new polyurethane material with excellent performance and environmental protection by changing the R value and film-forming temperature, which provides a valuable method for the sustainable utilization of renewable resources.

## 2. Experimental Methods

### 2.1. Materials

Isophorone diisocyanate (IPDI) (Aladdin, Shanghai, China) and Dibutyltin dilaurate (DBTDL) (Aladdin, Shanghai, China) was used without further purification. Polyethylene glycol (PEG, Mw = 600) (Aladdin, Shanghai, China) was dried in vacuum at 80 °C for 24 h. Epoxidized soybean oil (ESO) (Aladdin, Shanghai, China) was chemically pure. Sulfuric acid (H_2_SO_4_, 98 wt%) was provided by Tianjin University of Science and Technology. Except for epoxy soybean oil, all other reagents were analytically pure.

### 2.2. Empirical Section

#### 2.2.1. Preparation of Soybean Oil-Based Polyester Polyols

Figure 1a,b are the reaction device diagram and preparation route diagram of soybean oil polyester polyol. Polyethylene glycol and epoxy soybean oil were placed in a three-neck flask with a mechanical agitator and a condenser. Their molar ratio is 1.66:1. Under the catalysis of concentrated sulfuric acid, the soybean oil-based polyester polyols was obtained by reaction at 100 °C for 3 h. After the reaction, concentrated sulfuric acid was neutralized with sodium carbonate solution, and soybean oil-based polyester polyols were extracted with ethyl acetate. Finally, the ethyl acetate was removed by a rotary evaporator at 94 °C to obtain a faint yellow viscous liquid which was a soybean oil-based polyester polyol.

#### 2.2.2. Preparation of Soybean Oil-Based Polyurethane Thin Film

Isophorone diisocyanate (IPDI) and soybean oil polyester polyols were placed in a three-neck flask with a mechanical agitator and a condenser using a small amount of N, N dimethylformamide (DMF) as a solvent. Under the catalysis of DBTDL, the soybean oil-based polyurethane (SPU) prepolymer was obtained by reacting at 85 °C for 1 h. The prepared SPU was placed in a mold and dried in a vacuum drying oven for 8 h to obtain the film. The properties of SPU films were investigated by changing the R value (-NCO: -OH) and the film forming temperature.

During the experiment, we found that when the R value is less than 1.6, the polyol content in the polymer is too high, and the hydroxyl group in the molecule is dehydrated, which makes the molecular weight of the polymer decrease. Consequently, the viscosity of the sample is too high to form a film, as shown in Figure 2a. When the R value is greater than 2.0, the polymer molecules cross-link internally, forming a gel state during the synthesis reaction; the film cannot be formed, as shown in Figure 2b. When the film-forming temperature is lower than 40 °C, the sample obtained after drying is in the shape of wax sheet instead of film, as shown in Figure 2c. When the film temperature is higher than 80 °C, a large number of bubbles appear on the SPU film after drying, which may be caused by the rapid evaporation of the solvent, as shown in Figure 2d.

For the reasons mentioned above, we set the R value range from 1.6 to 2.0 and the film forming temperature range from 40 °C to 80 °C on the premise of the R value.

### 2.3. Characterization

FTIR spectra of soybean oil-based polyester polyols and polyurethane films were recorded with a Bruker-Vector 22 FTIR spectrophotometer (Billerica, MA, USA), within the range of 4000 cm^−1^–500 cm^−1^.

The ^1^H NMR spectra of soybean oil-based polyester polyols were studied using an A Bruker Avance 400 spectrometer (400 MHz).

The molecular weight and polydispersity index (PDI) of the soybean oil-based polyurethane were determined by gel permeation chromatography (GPC) utilizing a Waters model 515 pump and a model 2410 differential refractometer. Methyl trichloride was used as the eluent at a flow rate of 1.0 mL/min. Calibrated with polystyrene standard.

The SU-1510 scanning electron microscopy (SEM) was used to analyze the fracture morphology of the sample at the accelerating voltage of 10.0 kV.

The DSC 8000-TA instrument (New Castle, DE, USA) was used to perform differential scanning calorimetry. The temperature test range was −80 °C to 200 °C and the heating rate was 10 °C /min.

The thermal stability and thermal decomposition rate of the soybean oil-based polyurethane were analyzed by Q500 thermogravimetric analyzer produced by TA Instruments. The heating rate was 10 °C /min from 0 °C up to 600 °C under nitrogen gas.

The VCA Optima contact angle measuring instrument (KRUSS, Hamburg, Germany) was used to study the surface hydrophobicity of polyurethane films.

The UV-2700 spectrometer (Shimadzu, Kyoto, Japan) was used to measure the spectral permeation rate of the polyurethane films. The spectral range is 200 to 800 nm.

Instron 3369 universal testing machine was used to study the mechanical properties of polyurethane films according to GB/T 1040-2006.

GDP-C permeability tester (Brugger, Munich, Germany) was used to study the oxygen permeability of the polyurethane films. The samples were prepared according to the GB/T 1038-2000. Three parallel specimens were selected for each sample during the test. The transmittance *Pg* of the film was calculated by the formula.
Pg = 1.157 × Qg × D
wherein, *Pg* is the gas transmittance of the film, in g·cm(cm^2^·s·Pa)^−1^ and *Qg* is the amount of gas passing through of the film, in g(cm^2^·s·Pa)^−1^. *D* is the thickness of the specimen (in cm).

## 3. Results and Discussion

### 3.1. Characterization of Soybean Oil-Based Polyester Polyols Structure

The infrared spectra of epoxy soybean oil (ESO), polyethylene glycol and soybean oil-based polyester polyol are shown in Figure 3. The epoxy peak of epoxy soybean oil is at 1253 cm^−1^(8 μ), 823 cm^−1^(11 μ) and 724 cm^−1^(12 μ). The stretching vibration peak of primary alcohol is at 3429 cm^−1^–3464 cm^−1^. The C-O vibration peak of primary alcohol occurs at 1124 cm^−1^–1093 cm^−1^. The asymmetric and symmetrical stretching vibration caused by long-chain fatty acids -CH_2_ and -CH_3_ cause absorption peaks to appear at 2919 cm^−1^–2923 cm^−1^. In addition to the above, the FT-IR spectrum of soybean oil-based polyester polyol also shows the absorption peaks at 1731–1748 cm^−1^ attributed to C-O stretching vibration. The stretching vibration peak of C-O in ether is at 1100 cm^−1^–1200 cm^−1^. This confirms the reaction of PEG to epoxy.

The ^1^H NMR of ESO and soybean oil-based polyester polyol (ESO-polyol) are shown in Figure 4. Comparing these two figures, it can be seen that the characteristic shift double peaks of epoxy groups of ESO-Polyol at 2.8–3.2 ppm disappear, while the methylene peak of PEG chain appears at 3.6 ppm. The peaks in other positions remained basically, which showed that the main structure of ESO was not destroyed. The ^1^H NMR spectrum and chemical shifts are very consistent with the chemical structure of the polyester polyols.

GPC results showed that the number average molecular weight of ESO-polyol was 2815, the weight average molecular weight was 3009, and the polydispersity was 1.07.

### 3.2. Structure Characterization of Soybean Oil-Based Polyurethane Film

As shown in Figure 5, the FT-IR of SPU has no characteristic peak of -NCO at 2250 cm^−1^, which indicates that -NCO is fully involved in the reaction. There is no free N-H absorption peak near 3441 cm^−1^. 3335 cm^−1^ is the absorption peak of N-H produced by hydrogen bonding. The absorption peak at 1549 cm^−1^ is plane symmetric N-H in SPU. This shows that almost all of the hard segment N-H are involved in formation of hydrogen bonds. At 2856 cm^−1^ and 2924 cm^−1^ are the symmetric and antisymmetric stretching vibration peaks of -CH_2_, respectively. The strong peak at 1739 cm^−1^ is the stretching vibration peak of the urethane bond carbonyl group. The peak at 1549 cm^−1^ is the N-H in-plane symmetrical bending vibration absorption peak. At 1459 cm^−1^ is the -CH_2_ in-plane bending vibration peak. 1061cm^−1^ is the symmetrical stretching vibration peak of C-O-C in the CO-O-C of the hard segment chain. At 955 cm^−1^ and 1239 cm^−1^ are the symmetrical and anti-symmetrical stretching vibration peaks of O-C-O and C-O-C in the soft segment polyester, respectively. At 774 cm^−1^ is the deformation vibration peak of the urethane bond (O-CO-NH-). The appearance of these characteristic peaks indicates the synthesis of polyester polyurethane.

GPC result showed that the number average molecular weight of SPU was 30,276, the weight average molecular weight was 59,215, and the polydispersity was 1.96.

### 3.3. Performance Test of Soybean Oil-Based Polyurethane Films with Different R Values

#### 3.3.1. FT-IR Spectra of Soybean Bio-Based Polyurethane Films with Different R Values

Comparing Figure 6 and Table 1 with the analysis at 3.2, it can be seen that each SPU sample was successfully prepared. At 3330 cm^−1^, the peak intensity of the N-H stretching vibration absorption peak increases with the increase of the R value, and the peak gradually shifts to the low frequency. This is due to the fact that the increase in the R value increases the hard segment content, which will form more hydrogen bonds and physical cross-linking points [19]. The role of hydrogen bonds mainly occurs between the hard segments. The molecular chain is flexible and the molecular structure with small steric hindrance is conducive to the formation of hydrogen bonds [20].

#### 3.3.2. Scanning Electron Microscopic Analysis of Soybean Bio-Based Polyurethane Films with Different R Values

The self-assembly behavior of block copolymers is similar to that of amphiphilic small molecules. The solvent-compatible segments act together with the solvent-repellent segments to promote self-assembly of the block copolymers in some solvents, forming micelles with various morphological structures such as spherical, rod-shaped, vesicle-shaped, and fiber-shaped. The self-assembly morphology of the polyurethane synthesized in this experiment was shown in Figure 7. Among them, the white area represents the hard segment with higher cohesive energy in the polyurethane, forming the dispersed phase. The dark areas represent the soft segments in the polyurethane, forming a continuous phase. It can be seen that all of the SPU film samples can be observed in both light and dark phases, which confirms the phase separation structure of the soft and hard segments of the SPU films. It can also be observed that when the R value is 1.6, there are fewer white areas in the microscopic section of the SPU film, and the section is relatively smooth. When the R value is 2.0, the white areas in the microscopic section of SPU film obviously increases, and the sectional structure becomes rough and uneven. This indicates that the compatibility of SPU films becomes worse and the degree of microphase separation becomes larger with the increase of the R value.

#### 3.3.3. DSC Analysis of Soybean Bio-Based Polyurethane Films with Different R Values

The glass transition temperature (T_g_) of the soft segment (polyol-ESO) of the PU film can be obtained from the DSC analyzer, and the T_g_ can indicate the degree of microphase separation between the hard and soft segments. That is, the lower the T_g_ of the soft segment, the higher the microphase separation degree of the PU film. Furthermore, the higher the R value of SPU, the lower the T_g_ [21]. Figure 8 shows the effect of R value on the T_g_ of PU film.

As shown in Figure 8, the larger the R value, the lower the T_g_ of the PU film. Since with the increase of R value, the average length of hard segments and the aggregation between molecules increase, there is poor compatibility between the soft and hard segments and an increase of microphase separation. In addition, the curve in the figure has some less obvious endothermic melting peaks between −70 °C and 200 °C, which indicates that there are some small amounts of crystals in the PU film.

#### 3.3.4. Thermogravimetric Analysis of Soybean Bio-Based Polyurethane Films with Different R Values

Thermogravimetric analysis can analyze the mass change and thermal stability of SPU film. The TGA curve of SPU film was shown in Figure 9. Since the decomposition temperature of the soft segment is much higher than that of the hard segment, the SPU weight loss is divided into two stages, which are attributed to the decomposition of the hard segment and the soft segment, respectively. The weight loss in the first stage was about at 230 °C, and the mass of the polymer began to slowly decline. This is because the urethane bond of the rigid segment breaks to produce isocyanate, carbon dioxide, alcohol, primary amine and secondary amine [22,23]. In the process of continuous temperature rise, the rate of weight loss appears the maximum value. Thermal decomposition is carried out at about 470 °C to the soft segment. At this time, as the temperature rises, the thermal decomposition process becomes very slow and stops at around 600 °C. It can be seen from the figure that the temperature of SPU films with different R values at a weight loss of 2.5% is divided into 234.3 °C, 255.7 °C, 266.5 °C, 266.8 °C, and 280 °C, and the residual amount after heating to 470 °C is about 3%, indicating that SPU has good thermal stability. The higher the R value, the higher the thermal decomposition temperature. In other words, the higher the R value, the higher the heat resistance. As the R value increases, the number of polar groups in the molecular chain increases, the hydrogen bond becomes stronger, the intramolecular strength increases and the heat resistance increases.

#### 3.3.5. Analysis of Hydrophilic/Hydrophobic Properties of Soybean Bio-Based Polyurethane Films with Different R Values

Contact angle is an important measure to measure the surface hydrophilicity and hydrophobicity of materials. When the contact angle is less than 90°, the material is hydrophilic, and when the contact angle is greater than 90° it is hydrophobic. It can be seen from Figure 10 that the contact angles of all materials are greater than 90°, and with the increase of R value, the contact angles also increase (that is, the hydrophobicity of materials increases). This is due to the fact that the hard segment content increases with the increase of the R value, this will cause the intermolecular force to increase, the intermolecular space becomes smaller, the degree of crosslinking becomes larger, and it is more difficult for water molecules to enter. As the hard segment increases, the hydroxyl content will be relatively reduced, which is also the reason for the decrease in the hydrophilicity of the film surface. Moreover, the ester structure in the soybean oil-based polyester polyol is very hydrophobic.

#### 3.3.6. Analysis of Light Transmittance of Soybean Bio-Based Polyurethane Films with Different R Values

The light transmittance of the film is related to the degree of crystallization. The crystallization behavior will cause the incident light to deviate from the incident direction and affect the degree of transmission of it in the film. It can be seen from Figure 11 that the transmittance of the film is in the range of 30–42%, and the transmittance of the film gradually decreases with the increase of the R value. When the R value increases to a certain extent, the molecular chain will be fully regular and crystallized. The crystallization of molecular chains has a great influence on the light transmittance of the film [24,25]. In addition, if the surface of the film is not smooth and uniform, it may cause light scattering, deviating from the direction of incident light and thus reducing the light transmittance.

#### 3.3.7. Analysis of Mechanical Properties of Soybean Bio-Based Polyurethane Films with Different R Values

Figure 12 shows the effect of R value on the mechanical properties of SPU films. With the increase of R value, the elongation at break shows a downward trend, and the tensile strength increases with the increase of R value. As the R value increases, the hard segment content also increases, which makes the relative slip between the molecular chains difficult. At the same time, this also reduces the content of the soft segment aggregation zone, and the force between the molecules of the hard segment increases, which leads to an increase in the tensile strength of the film and a decrease in the elongation at break. When the film is subjected to external force, the range of space that the molecular segments can move between becomes smaller and it is difficult to deform, so the elongation at break decreases.

#### 3.3.8. Analysis of Oxygen Permeability of Soybean Bio-Based Polyurethane Films with Different R Values

Figure 13 shows the effect of different R values on the oxygen barrier properties of SPU films. As shown in the Figure 13, the higher the R value, the lower the oxygen permeability coefficient of the SPU film. Especially when R is 2.0, the oxygen permeability coefficient of the SPU film is reduced to 0.27 × 10^−14^ cm^3^·cm/(cm^2^·s·pa). The aggregation state and chemical structure of the polymer are the main factors affecting air permeability. According to the DSC and transparency test results, the R value increases, the number of polar groups-NCO increases, the crystallinity of the molecular chain increases, the amorphous area decreases, the tightness of the molecular chain arrangement inside the SPU film increases, and the small molecules increase at the same time. The complexity of the gas path makes it difficult for small molecule gases to permeate and reduces the oxygen permeability of the SPU film.

### 3.4. Performance Test of Soybean Oil-Based Polyurethane Films at Different Filming Temperatures

#### 3.4.1. FT-IR Spectra of Soybean Bio-Based Polyurethane Films at Different Film Forming Temperatures

Figure 14a is the infrared spectra of the N-H bonds of SPU films at different film forming temperatures. It can be seen that no free NH stretching vibration peak is observed at 3400 cm^−1^–3600 cm^−1^, only the hydrogen bonding NH stretching vibration peak at 3322 cm^−1^ appears, indicating that almost all hard segment NH is involved at the same time, as the film-forming temperature increases, the absorption peak intensity of the NH bond gradually increases, indicating that as the film-forming temperature increases, the hydrogen bonding between the hard segments gradually increases.

As shown in the infrared spectrum of SPU in Figure 14b, the characteristic peak of -NCO group at 2250 cm^−1^ disappears, -NCO fully participates in the reaction, and there is no residue of -NCO group in the synthesized product.

#### 3.4.2. Scanning Electron Microscopy Analysis of Soybean Bio-Based Polyurethane Films at Different Film Forming Temperatures

As shown in Figure 15, when the film forming temperature is 40 °C, there are fewer white areas in the microscopic section of the SPU film, and the section is relatively smooth, which indicates that the SPU film has better compatibility between the soft and hard segments and the degree of microphase separation is smaller. When the film forming temperature is 80 °C, the white areas in the microscopic section of SPU film obviously increases. It may be that the aggregation degree of hard segments in the molecular structure of SPU increases when the film is cured at 80 °C, which makes the microphase separation degree larger. Thus, the cross-sectional structure of SPU film becomes rough and uneven. In addition, as the film-forming temperature increases, the white areas in the microscopic section of the SPU film gradually increase and the section becomes more and more rough, indicating that as the film forming temperature increases, the compatibility of the SPU film becomes worse, and the degree of microphase separation increases.

#### 3.4.3. Analysis of Hydrophilic/Hydrophobic Properties of Soybean Bio-Based Polyurethane Films at Different Film Forming Temperatures

Figure 16 shows the effect of film-forming temperature on the water contact angle of bio-based polyurethane films. With the increase of the film-forming temperature, the water contact angle of the SPU film gradually increases. When the film-forming temperature is 80 °C, the water contact angle of the SPU film is up to 93.7°. The reason for this phenomenon may be that at lower film forming temperature, the SPU film surface becomes slightly sticky, resulting in a slightly smaller contact angle. When the film-forming temperature is 80 °C, the surface of the SPU film is slightly sticky. The degree of smoothness is better, the molecular arrangement becomes tighter, the gap between the molecules is reduced, and it is difficult for water molecules to enter. Thus, the contact angle is larger and the hydrophobicity improves.

#### 3.4.4. Analysis of Light Transmittance of Soybean Bio-Based Polyurethane Films with Different Film Forming Temperatures

Figure 17 shows the effect of film forming temperature on the transparency of SPU films. As the film-forming temperature increases, the transparency of the SPU film gradually decreases. The main reason for this is that the film forming temperature increases, the order of the hard segment of the SPU film increases, the molecular bond hydrogen bonding of the hard segment increases, the degree of microphase separation of the soft and hard segments becomes greater, and the crystallization behavior of the hard segment microdomains and the crystallinity increase. The crystallinity is not conducive to the transmission of light, which causes the transparency of the SPU film to decrease.

#### 3.4.5. Analysis of Mechanical Properties of Soybean Bio-Based Polyurethane Films with Different Film Forming Temperatures

Figure 18 shows the effect of film forming temperature on the mechanical properties of SPU films. As shown in Figure 18, as the film-forming temperature increases, the tensile strength of the SPU film gradually increases, reaching a maximum of 6.04 Mpa when the film-forming temperature is 80 °C; however, the breaking elongation of the SPU film rate decreases with the increase of the film-forming temperature, from 414.794% when the film-forming temperature is 40 °C to 301.3724% when the film-forming temperature is 80 °C.

As the film forming temperature increases, the tensile strength of the SPU film shows an upward trend. This is due to the fact the movement of the molecular chain of the SPU intensifies after the temperature rises, which makes the arrangement of the molecular chain of the SPU very tight, and the hydrogen bonding effect between the molecular chains will also increase with the hard segment chain. The degree of aggregation increases and strengthens. The force between the molecules of the rigid segment increases, which increases the tensile strength of the film and reduces the elongation at break. When an external force is applied to the film, the movable space range of the molecular chain is reduced, and it is uneasy to deform, so the elongation at break is reduced.

#### 3.4.6. Oxygen Permeability Analysis of Soybean Bio-Based Polyurethane Films with Different Film Forming Temperatures

Figure 19 shows the effect of film-forming temperature on the oxygen permeability of SPU films. The oxygen permeability coefficient of SPU films showed a decreasing trend with the increase of film forming temperature. When the film-forming temperature is 80 °C, the oxygen permeability coefficient of the SPU film is 0.203 × 10^−14^ cm^3^·cm/(cm^2^·s·pa). As the film-forming temperature rises, the crystallinity increases, the amorphous area in the molecular structure decreases, the tightness of the molecular chain arrangement inside the film increases, and the gap between the molecules becomes smaller. Therefore, the gap for oxygen permeation becomes smaller, resulting in permeability. Consequently, the oxygen permeability coefficient drops.

#### 3.4.7. Thermogravimetric Analysis of Soybean Bio-Based Polyurethane Films with Different Film Forming Temperatures

Figure 20 shows the thermogravimetric curves of SPU film cured at 60 °C and 80 °C. From the thermogravimetric loss curve in Figure 20, we can see that the SPU decomposition trends at these two temperatures are similar. Its temperature has no apparent effect on the thermal resistance of the SPU. Specific degradation peaks for all samples are shown in the curves obtained from the Figure 20 weight loss DTG.

The DTG curve has two major peaks. The first is the initial weak degradation peak at about 330 °C. You can see that the first main decomposition peak has two extremum points. Therefore, the main decomposition peak is considered to be caused by two different peaks. The second is the main decomposition peak at about 420 °C, so the main decomposition peak is considered to consist of two different peaks. Therefore, there are three main decay peaks. The first stage with a weight loss in the range of 4–7% by weight can be considered as the residual organic solvent of the SPU film. The main decomposition peak near 230 °C to 420 °C may be the decomposition of the hard and soft segments of the SPU. The degradation peak from about 230 °C could be due to the rigid SPU segment, and its weight loss is consistent with the weight ratio of the rigid PU segment. Polyurethane is a block copolymer with hard and soft segments. The inflection of the DTG curve can be caused by a domain different from the polyurethane phase separation [26,27,28]. Note that the maximum weight loss temperature of the film curing SPU at 60 °C during this decomposition step is 421.47 °C and the maximum weight loss temperature at 80 °C is 423.58 °C, showing a slight drop. It is important to do. Hard segment comparison. It can be inferred that this is caused by the number of hydrogen bonds in the polyurethane caused by temperature [29,30]. The rise in temperature promotes the formation of hydrogen bonds.

## 4. Conclusions

In this study, epoxy soybean oil (ESO) was used as raw material and polyethylene glycol (PEG) was used for ring opening reaction. FT-IR and ^1^H NMR tests proved that soybean oil-based polyester polyol was successfully prepared. Then soybean oil-based polyurethane (SPU) was synthesized by the reaction of this bio-based polyol with isophorone diisocyanate (IPDI). The properties of SPU films were studied by changing R value and film forming temperature.

The R value is set from 1.6 to 2.0. The experimental results show that the tensile strength of SPU film increases from 5.18 MPa to 12.58 MPa with the increase of R value. In the range of R value, all contact angles are greater than 90°, which is hydrophobic, and increasing R value will increase the water contact angle of the film. The heat resistance of SPU thin film is improved, and the temperature at 2.5% weight loss increases from 234.3 °C to 280 °C. The higher the R value, the lower the oxygen permeability coefficient and moisture permeability coefficient of SPU film. Especially when R value is 2.0, the oxygen permeability coefficient of SPU film decreases to 0.27 × 10^−14^ cm^3^·cm/(cm^2^·s·pa), and the moisture permeability coefficient is 1.67 × 10^−14^ cm^3^·cm/ (cm^2^·s·pa), which improves the barrier property. However, with the increase of R value, the transmittance of the film gradually decreases.

The setting range of film forming temperature is 40 °C to 80 °C. When the film-forming temperature increases from 40 °C to 80 °C, the absorption peak strength of the N-H bond of SPU increases gradually, and the hydrogen bond between hard segments increases gradually. The tensile strength of SPU films increased from 3.25 MPa to 6.04 MPa, and the water contact angle increased from 87.70° to 93.70°. When the film temperature is 80 °C, the oxygen permeability coefficient of SPU film is 0.203 × 10^−14^ cm^3^·cm/(cm^2^·s·pa), and the moisture permeability coefficient is 4.05 × 10^−14^ cm^3^·cm/(cm^2^·s·pa). The mechanical properties, hydrophobicity, thermal stability and barrier properties of SPU films are effectively improved. However, with the increase of film-forming temperature, the transparency of the film gradually decreases.

If SPU film is used in packaging field, considering the thermal stability, mechanical properties, barrier properties and the transparency of the film, it is considered that the best effect is when the R value is 1.8 and the film forming temperature is 60 °C. This study successfully provides a valuable method for the sustainable utilization of renewable resources.

## Figures and Tables

**Figure 1 polymers-14-02201-f001:**
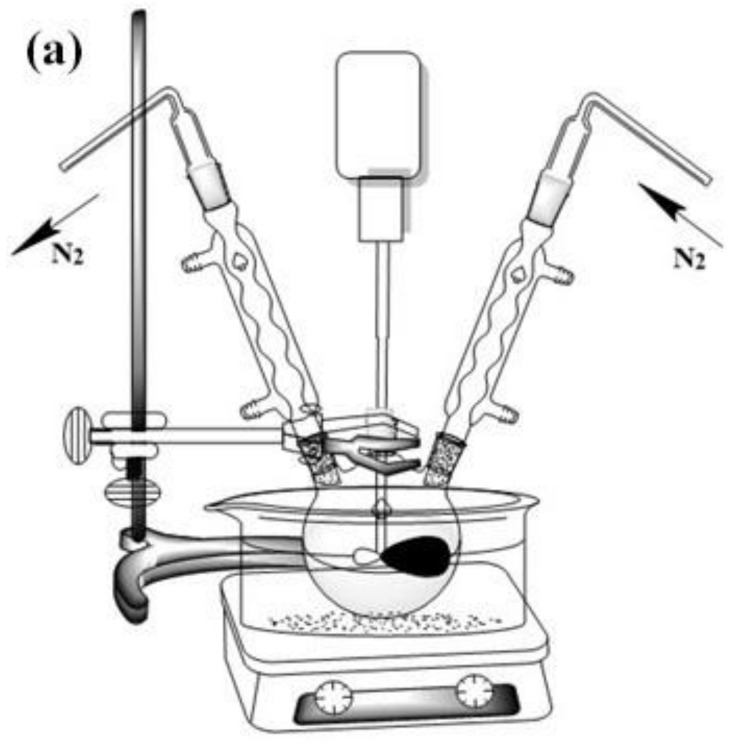
(**a**) Soybean oil-based polyester polyol reaction device; (**b**) soybean oil-based polyester polyol reaction principle.

**Figure 2 polymers-14-02201-f002:**
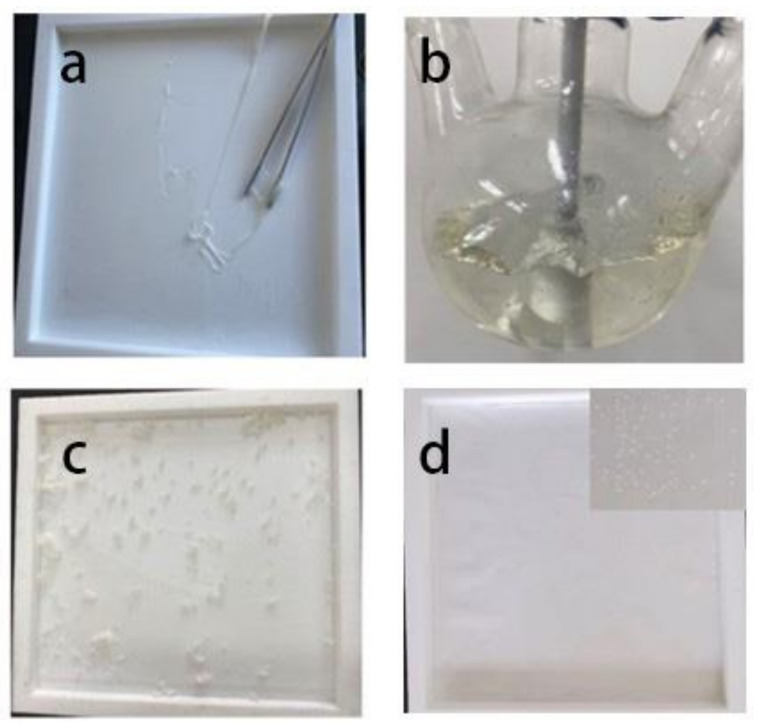
(**a**) When the R value is too small, the BPU cannot form a film. (**b**) When the R value is too large, the BPU in the gel state. (**c**) When the film forming temperature is low, the BPU in the wax form. (**d**) When the film forming temperature is too high, a large number of bubbles appear in BPU.

**Figure 3 polymers-14-02201-f003:**
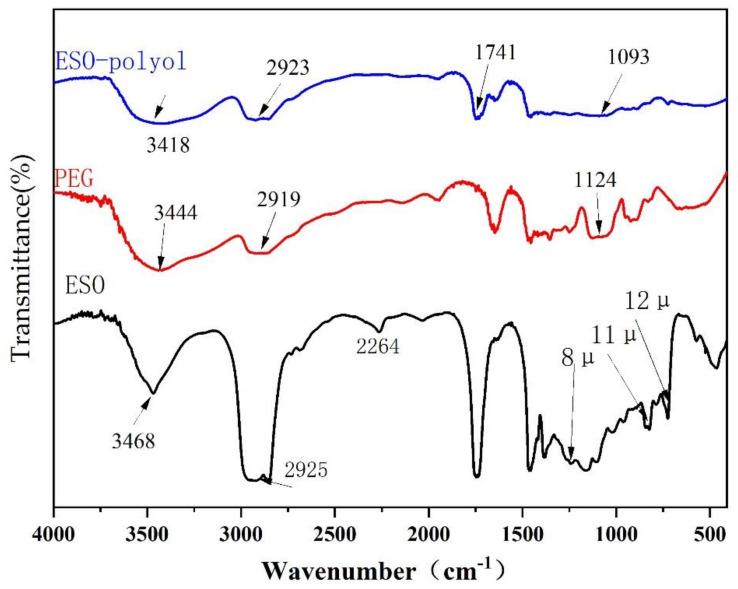
Comparison of epoxidized soybean oil, polyol, and soybean oil-based polyester polyol FT-IR spectrum.

**Figure 4 polymers-14-02201-f004:**
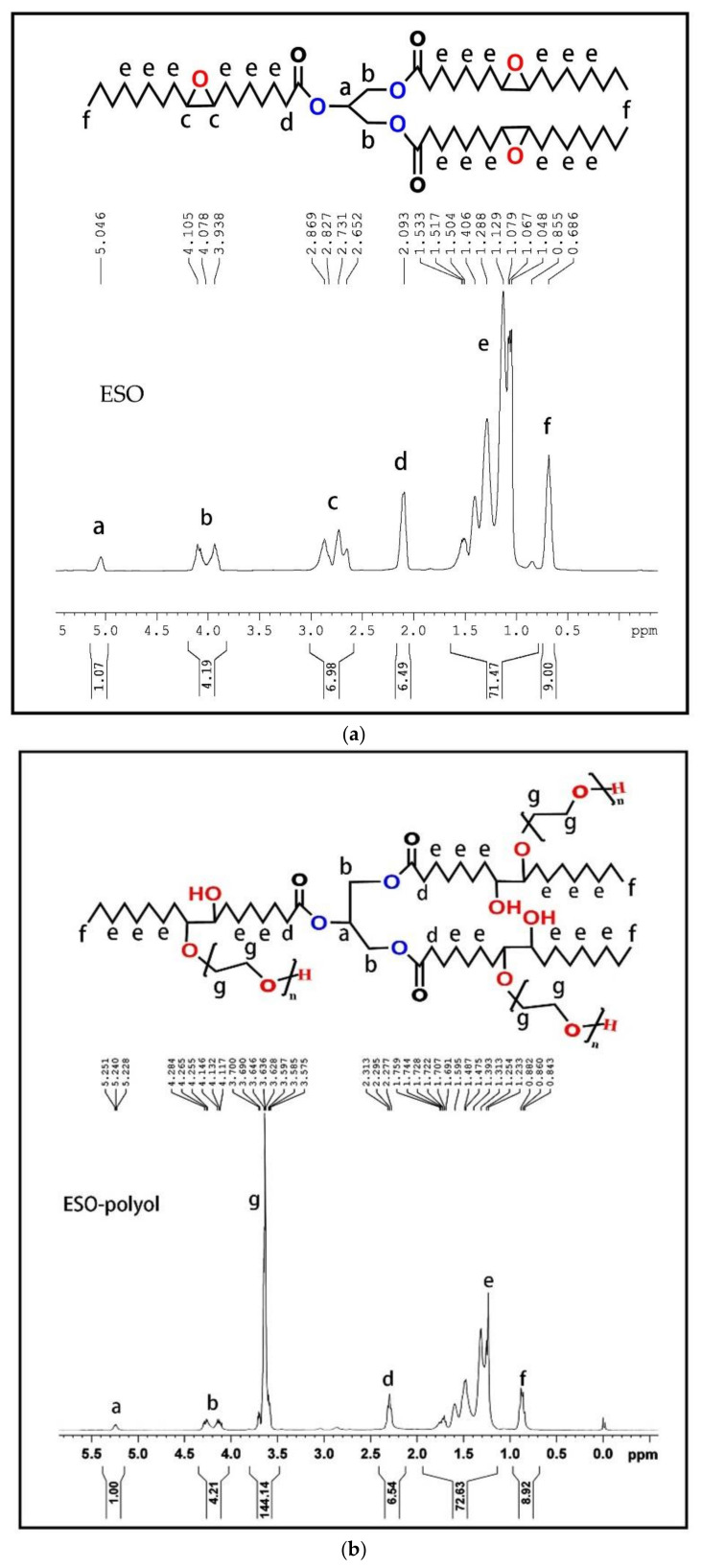
(**a**) ^1^H NMR spectra of ESO, (**b**) ^1^H NMR spectra of ESO-polyol.

**Figure 5 polymers-14-02201-f005:**
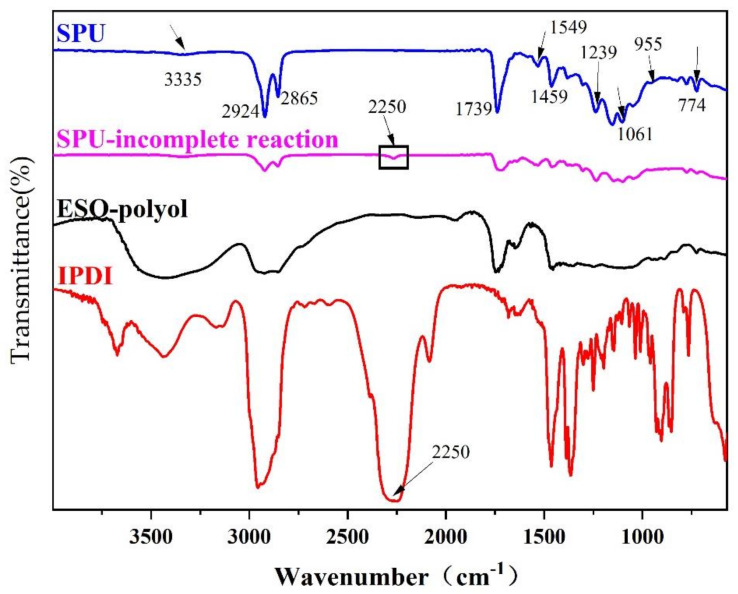
FT-IR spectrum of IPDI, Polyol-ESO, SPU, and SPU-incomplete reaction.

**Figure 6 polymers-14-02201-f006:**
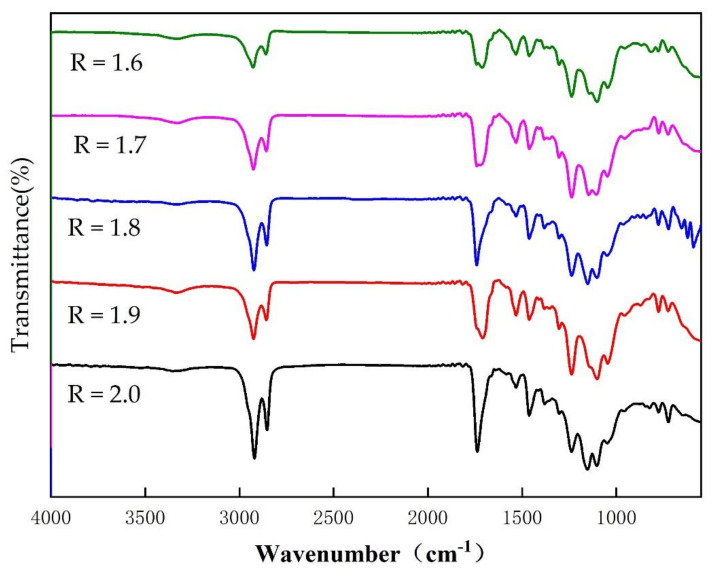
FT-IR spectra of SPU films with different R values.

**Figure 7 polymers-14-02201-f007:**
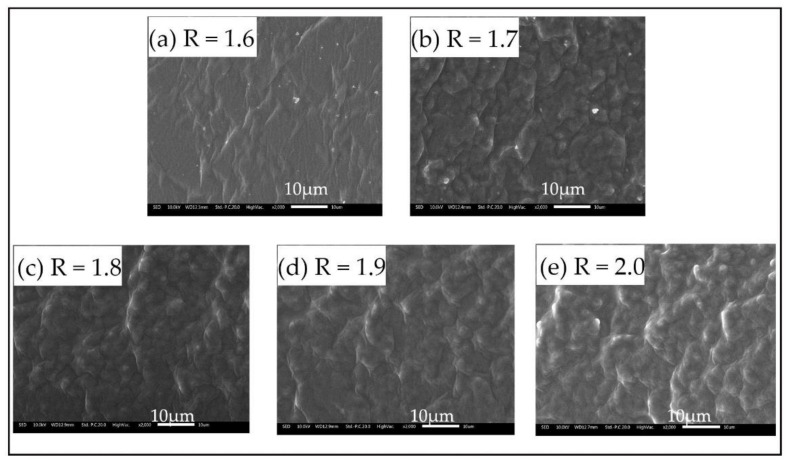
Scanning electron microscope cross-sections of SPU films with different R values. (**a**) R =1.6, (**b**) R = 1.7, (**c**) R = 1.8, (**d**) R = 1.9, (**e**) R = 2.0.

**Figure 8 polymers-14-02201-f008:**
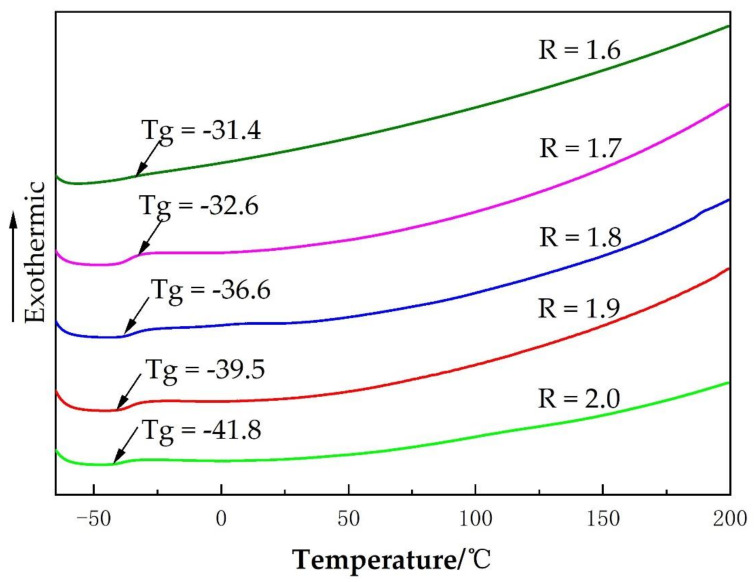
DSC curves of SPU films with different R values.

**Figure 9 polymers-14-02201-f009:**
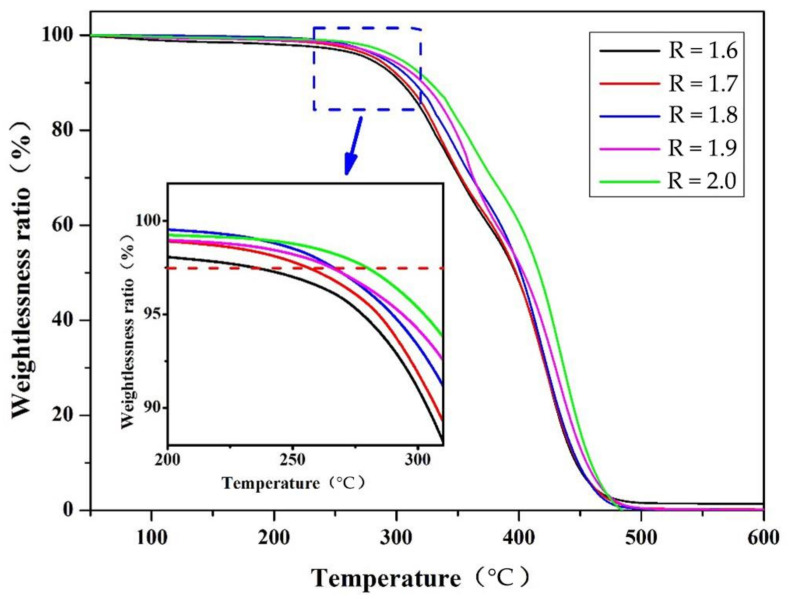
TGA curves of SPU films with different R values.

**Figure 10 polymers-14-02201-f010:**
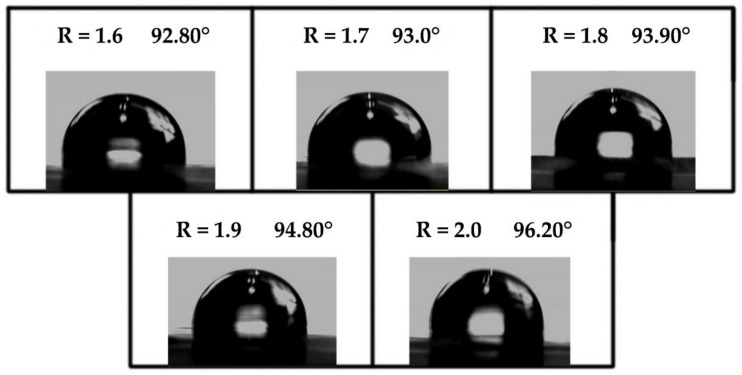
Water contact angles of SPU films with different R values.

**Figure 11 polymers-14-02201-f011:**
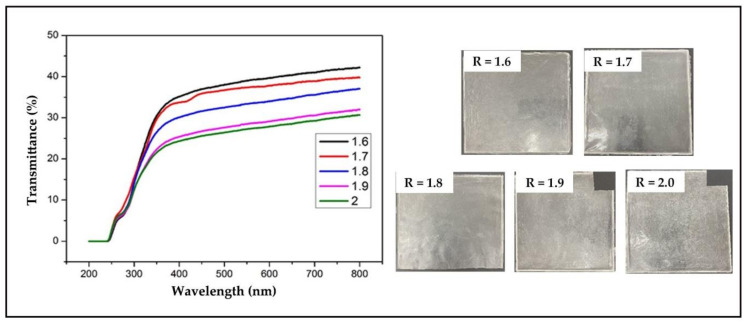
UV transmittance spectrum and photos of SPU films with different R values.

**Figure 12 polymers-14-02201-f012:**
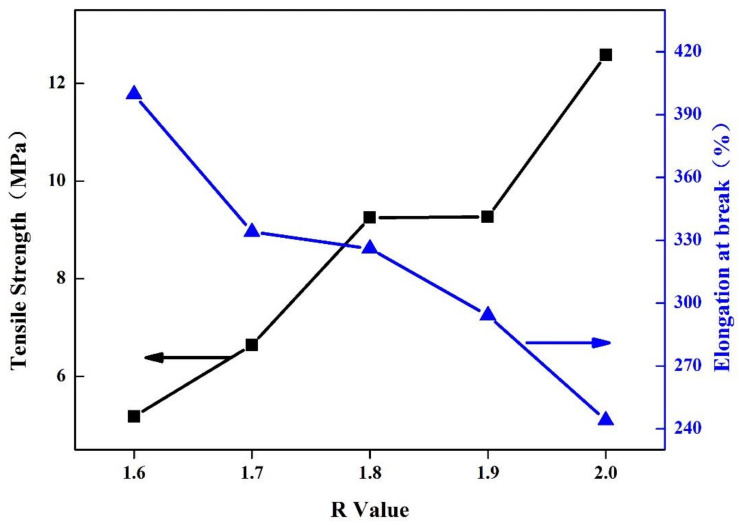
Mechanical properties of SPU films with different R values.

**Figure 13 polymers-14-02201-f013:**
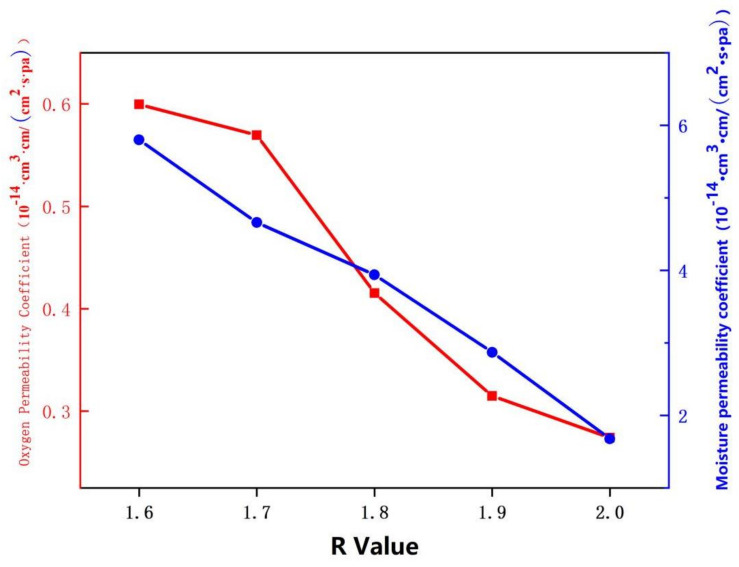
Oxygen permeability of SPU films with different R values.

**Figure 14 polymers-14-02201-f014:**
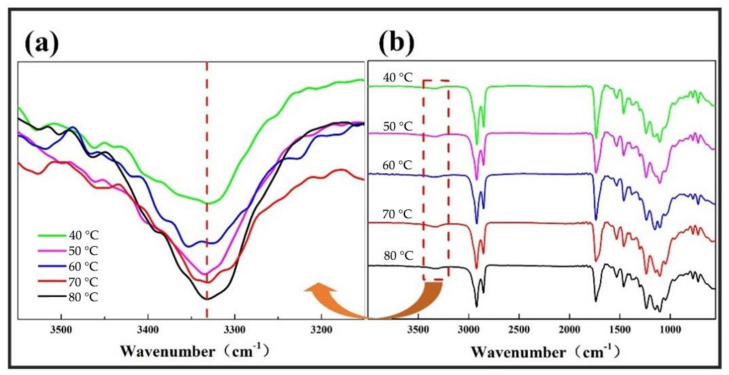
(**a**) FT-IR spectra of the N-H bonds of SPU films with different film forming temperatures, (**b**) FT-IR spectra of SPU films with different film forming temperatures.

**Figure 15 polymers-14-02201-f015:**
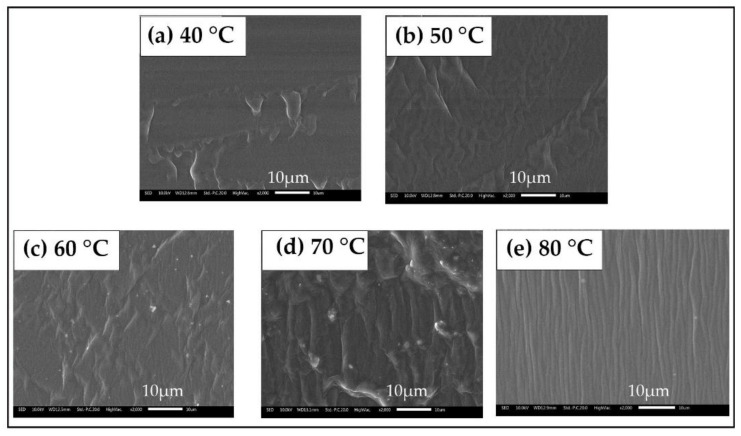
Scanning electron microscope cross-sections of SPU films at film forming temperatures of (**a**) 40 °C, (**b**) 50 °C, (**c**) 60 °C, (**d**) 70 °C, (**e**) 80 °C.

**Figure 16 polymers-14-02201-f016:**
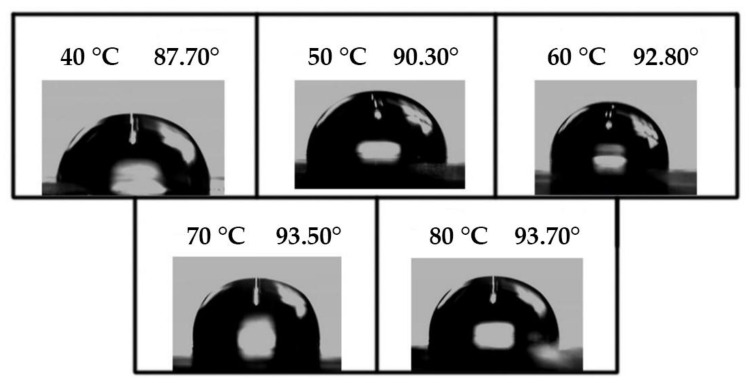
Water contact angles of SPU films with different film forming temperatures.

**Figure 17 polymers-14-02201-f017:**
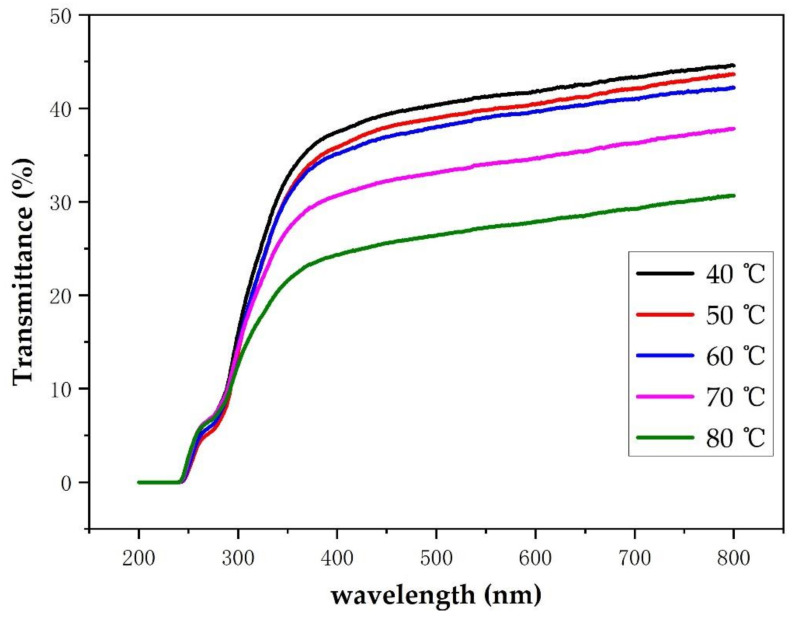
UV transmittance spectrum and photos of SPU films with different film forming temperatures.

**Figure 18 polymers-14-02201-f018:**
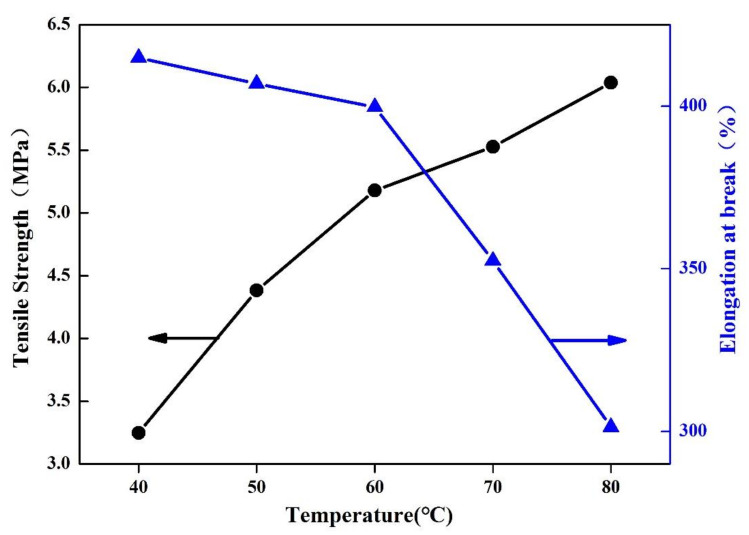
Mechanical properties of SPU films with different film forming temperatures.

**Figure 19 polymers-14-02201-f019:**
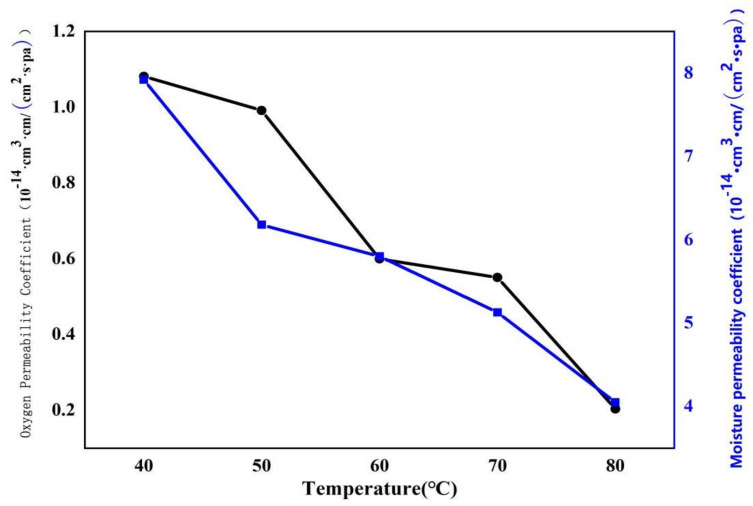
Oxygen permeability of SPU films with different film forming temperatures.

**Figure 20 polymers-14-02201-f020:**
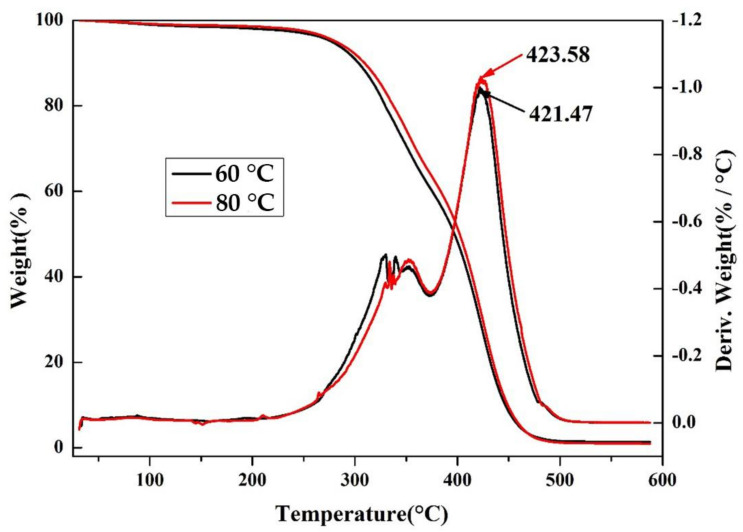
TGA curves of SPU films with different film forming temperatures.

**Table 1 polymers-14-02201-t001:** Displacement of vibration peaks in bio-based polyurethane films with different R value.

Vibration Group	R = 1.6	R = 1.7	R = 1.8	R = 1.9	R = 2.0
-NH/cm^−1^	3350	3335	3330	3329	3326

## Data Availability

The authors declare data availability.

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
