# Peer review of "Synthesis, Characterization and Properties of Soybean Oil-Based Polyurethane"

_polymers, 2022, doi:10.3390/polym14112201_

Round 1
Reviewer 1 Report
This manuscript describes about soybean oil-based polyurethane. Obtained results are interesting, but I think some revision should be done.
1. There are much typing errors. Check and collect of the manuscript should be done. The examples are shown in as follows, but I think checking of all manuscript should be needed.
Line 12: ring-opening reaction .Soybean → ring-opening reaction. Soybean
Line 39: 13, 14 should be in parentheses.
Line 49: the Arabic numerals of “H2SO4” should be subscript.
Line 301: “-1” of 3322cm-1 should be superscript.
2. BPU (lines 17 and 159, and Figure 4) seems to be SPU.
3. What is the Y-axis of Figure 2?
4. At lines 132-133, the NMR signal seems not to be “absorption”.
5. In figure 3, where the signal based on epoxide exists?
6. How much is molecular weight of ESO-polyol?
7. At lines 160-162, the detail of the molecular weight of SPU such as Mw and Mn should be written . "Polydispersity of 1-2" is bad expression.
8. At Figure 5-12, the best value of R for the properties of SPU seems not to be observed. What do you think about the properties of SPU with R value at less than 1.5 or more than 2.0?
9. At section 3.4, what the R value of SPU was used? I think it should be written.
10. At Figure 13-18, the best foaming temperature for the properties of SPU seems not to be observed. What do you think about the properties of SPU at foaming temperature lower than 40℃ or higher than 80℃?
11. At Figure 16, usually, it is thought that a difference of 3-5 % of transmittance seems to be the measurement error in the solid sample. What do you think about this?
Author Response
Please see the details in the attachment

Reviewer 2 Report
The manuscript should be revised by a native English speaker or a professional language editing service to improve the grammar and readability.
Replace "petroleum resources, Oil is a non-renewable resource" line 33 by "non-renewable petroleum resources"
Correct "groups13, 14" line 39 to groups [13, 14].
The manuscript aim and abstract did not show the novelty of the submitted manuscript. Rewrite both and declare more the novelty in this manuscript.
All details of materials used (%purity, concentration, country of manufacturing and company of purchase) should be included.
All details concerning instruments used for analysis (model, country of manufacturing and company of purchase) should be included.
Mention the temperature used for removing ethylacetate using rotary evaporator.
Rewrite sections 2.2.1. and 2.2.2. to be more understandable. It is mainly composed of intermittent and interrupt sentences.
All instruments abbreviations should come after writing its whole name.
What do you mean by telescopic vibration lines 119, 121, 124, and 125?
The discussion of section 3.3.1 needs to be enriched and add references confirming your data interpretation.
What do you mean by "solvent-friendly" line 179?
Rewrite the data interpretation and discussion of section 3.3.2. what is written does not indicate anything.
What do you mean by Tg in section 3.3.3.? write the whole word before the abbreviation.
In figure 14 explain the obvious observed change in surface morphology at 80 C.
The discussion should be enriched and assured by valuable published references.
The conclusion seems to be summary of the performed work. Rewrite it to show the conclusion you reached from the obtained results and your recommendation based on interpretation of obtained results.
Round 2
Reviewer 1 Report
Some of the revision seems to be sufficient, but some of them should be checked.
- Typing errors still remain. The examples are shown in as follows, but I think checking of all manuscript should be needed. When you revised the manuscript, carefully check and collect of the manuscript should be done again.
At abstract, proton NMR was written as “1H NMR”, but at line 152 and 157, written as “1HNMR” (without spacing).
At line 61, “H2SO4,98wt%” seems to be “H2SO4, 98wt%” (with spacing).
- Why did you think information of the molecular weight of polyol-ESO is not important? I think this is very important information. If you want to show molecular weight of SPU, the information of the molecular weight of polyol-ESO should be written. Polyol-ESO is soluble because NMR spectra was indicated, therefore, I think measurement of molecular weight of poly-ESO is possible.
- At lines 182-183, even if the format was referred, I think “weight–average molecular weight of 52,000-59,000 and polydispersity of 1-2." is bad expression. If you want to show molecular weight of SPU, Mn and Mw (or Mw/Mn) should be shown. SPU seem to be soluble, therefore, I think measurement of Mn and Mw of SPU is possible.
Reviewer 2 Report
Thanks for doing all required corrections
Author Response
We checked the grammar and spelling of the whole article.
Round 3
Reviewer 1 Report
Thank you for revising. The manuscript has been revised satisfactorily.